MicroRNA-154-5p suppresses cervical carcinoma growth and metastasis by silencing Cullin2 in vitro and in vivo

Li Yaqin 1 2
Wei Yimiao 1
Zhang Honglei 3
Bai Ying 1
Wang Xiuting 4
Li Qi 1
Liu Yatao 1
Wang Shuling 5
Wang Jiapu 6
Wen Songquan 1
Li Jiarong 5
Zhao Weihong 1 sydeyzwh@sxmu.edu.cn
1 Department of Obstetrics and Gynecology, The Second Hospital of Shanxi Medical University , Taiyuan, Shanxi Province , China
2 Department of Obstetrics and Gynecology, Peking University People’s Hospital , Beijing , China
3 Department of Pathology and Pathophysiology,Basic Medical College, Shanxi Medical University , Taiyuan, Shanxi Province , China
4 Department of Biochemistry and Molecular Biology,Basic Medical College, Shanxi Medical University , Taiyuan, Shanxi Province , China
5 Department of Epidemiology,School of Public Health, Shanxi Medical University , Taiyuan, Shanxi Province , China
6 Scientific Research Experiment Center, Central laboratory, The Affiliated Cardiovascular Hospital of Shanxi Medical University , Taiyuan, Shanxi Province , China
Panda Amaresh
Electronic publication date: 2023 Jun 27
Publication date: 2023
Volume: 11
Electronic Location ID: e15641
Received 2023 Feb 21; Accepted 2023 Jun 5
Copyright: © 2023 Li et al.
Copyright year: 2023
Copyright holder: Li et al.
License: This is an open access article distributed under the terms of the Creative Commons Attribution License, which permits unrestricted use, distribution, reproduction and adaptation in any medium and for any purpose provided that it is properly attributed. For attribution, the original author(s), title, publication source (PeerJ) and either DOI or URL of the article must be cited.
License URL: https://creativecommons.org/licenses/by/4.0/

Keywords: Cervical cancer, MicroRNA-154-5p, Cullin2, Growth, Metastasis

Funding: National Natural Science Foundation of China 81702583 Outstanding Youth Fund Project of Shanxi Province 201901D211506 China Postdoctoral Science Foundation 2019M651072 Research Project Supported by Shanxi Scholarship Council of China 2022-195 This research was funded by the National Natural Science Foundation of China (Grant no. 81702583), the Outstanding Youth Fund Project of Shanxi Province (Grant no. 201901D211506), the China Postdoctoral Science Foundation (Grant no. 2019M651072) and the Research Project Supported by Shanxi Scholarship Council of China (Grant no. 2022-195). The funders had no role in study design, data collection and analysis, decision to publish, or preparation of the manuscript.

==============================
Background

MicroRNA-154-5p (miR-154-5p) plays a role in tumorigenesis in diverse human malignancies. Nevertheless, little is known about the mechanism by which miR-154-5p alters the growth and metastasis of cervical cancer. This research aimed to analyze the role of miR-154-5p in the pathology of cervical cancer in vitro and in vivo.

Methods

The level of miR-154-5p in human papillomavirus 16 positive cervical cancer cells was examined by real-time quantitative polymerase chain reaction. Bioinformatics predicted the downstream targets and potential functions of miR-154-5p. Furthermore, lentiviral technology was used to construct SiHa cell lines with stable up- and down-expression levels of miR-154-5p. Its differential expression effects on the progress and metastasis of cervical cancer were analyzed using cell culture and animal models.

Results

MiR-154-5p showed low expression in cervical cancer cells. Overexpression of miR-154-5p could markedly inhibit the proliferation, migration, and colony formation ability of SiHa cells, concomitantly leading to G1 arrest of the cell cycle, while silencing miR-154-5p triggered the opposite results. Meanwhile, overexpression of miR-154-5p restrained the growth and metastasis of cervical cancer by silencing CUL2 in vivo. Additionally, miR-154-5p reduced CUL2 level, and overexpression of CUL2 influenced the effect of miR-154-5p in cervical cancer. In conclusion, miR-154-5p restrained the growth and metastasis of cervical cancer by directly silencing CUL2.

Introduction

Cervical cancer is the fourth most common malignant tumor affecting women worldwide (Adiga et al., 2021). Approximately 604,000 new cervical cancer cases, along with 342,000 deaths, were registered globally in 2020 (Sung et al., 2021). Persistent high-risk human papillomavirus (HPV) infection acts as the leading cause of cervical cancer. Among them, HPV16 is the most frequent HPV genotype worldwide, accounting for around 50% of cervical cancer cases (Ferrall et al., 2021; Jiang et al., 2018). Therefore, blocking HPV16-mediated persistent infection is expected to interfere with the progression of cervical cancer.

The Ring E3 family are the most abundant types of ubiquitin ligases and key enzymes that are responsible for recruiting substrate proteins for degradation. Of them, Cullin2 (CUL2) provides the hydrophobin backbone for E3 ubiquitin ligase and participates in the formation of the CUL2 ubiquitin ligase complex (Shafique et al., 2018). Huh et al. (2007) demonstrated that the HPV16 E7/pRb complex contained CUL2, and knock-down of CUL2 using RNA interference led to an increase in the stability of pRb in HPV16 E7-expressed cells. Furthermore, abnormal HPV16 E7 and CUL2 complex expression resulted in pRb ubiquitination in vivo (Westrich et al., 2018). It is clear that CUL2 plays a critical role in the process of HPV16-induced cervical carcinoma.

MicroRNAs (miRNAs) are a category of endogenous, non-coding, single-stranded small RNAs with high conservatism, containing 21–23 nucleotides, which control tumorigenesis by regulating cancer-related functional genes or targeting cell signaling pathways (Nazarizadeh et al., 2020; Ai et al., 2022; Do et al., 2021). Related data revealed that miRNAs could affect the replication of high-risk HPV DNA, thereby regulating the life process of HPV and the mechanism of HPV-induced tumorigenesis (Li et al., 2021a). At the molecular-level, miRNAs are essential post-transcriptional gene regulators that target mRNA through the principle of sequence complementarity, which can directly inhibit translation or lead to mRNA instability, thereby degrading target proteins (Kadhim et al., 2023; Chatterjee et al., 2023; Zhao et al., 2022). In addition, key molecules that affect miRNAs biogenesis are located in complementary regions of target mRNA under miRNAs induction, leading to mRNA silencing (Shi et al., 2020).

MicroRNA-154-5p (miR-154-5p), an emerging miRNA, has low expression in various malignant tumors and exerts a tumor suppressor effect (Song et al., 2018; Tian et al., 2020; Chi et al., 2019). Previously, we selected miR-154-5p with the strongest ability to target and bind to CUL2 via tissue chip screening technology. The dual luciferase reporter assay confirmed the targeted regulation effect of miR-154-5p and CUL2, and preliminarily discussed the anti-cancer effect role of miR-154-5p mediated CUL2 in HPV16-positive cervical cancer cells using transient transfection method (Zhao et al., 2020). However, whether miR-154-5p can modulate the development of cervical cancer in vivo remains unknown. Therefore, this study chose miR-154-5p as the “main variable molecule,” and conducted both in vivo and in vitro experiments to clarify the effects of miR-154-5p targeting CUL2 on the evolution and metastasis of cervical cancer and identify new molecular targets for HPV16-induced cervical cancer.

Materials and Methods

Cell lines and laboratory animals

Human immortalized epidermal cell lines (HaCaT), 293T, and HPV16-positive human cervical cancer cell lines (SiHa) were acquired from the Cell Center of Shanghai Institutes for Biological Sciences (Shanghai, China), which were kept in Dulbecco’s modified Eagle’s medium (DMEM, Hyclone, Logan, UT, USA) supplemented with 10% fetal bovine serum (FBS) (Gibco, Gaithersburg, MD, USA) and 1% penicillin mixture (Solarbio, Beijing, China) in a moisturizing condition involving 5% CO2 at 37 °C.

A total of 32 female BALB/c nude mice (4–6 weeks) were obtained from Charles River Laboratories (Beijing, China). Groups were divided using random number table method. All nude mice were housed with 12 h light-dark cycles and stored in a specific pathogen-free (SPF) condition. The bedding was changed every 2 days, the room temperature was controlled at 22~26 °C, and the mice were fed with a feed rich in 18~20% protein. The entire experimental process followed the relevant regulations of the Ethics Committee of the Second Hospital of Shanxi Medical University on the welfare ethics of animal experiments (Approval No. DW2022035).

Lentiviral construction and cell grouping

All lentiviral vectors in this study carried green fluorescent protein (GFP) tags, and the construction scheme was completed by Fubio Biotechnology Co., Ltd (Suzhou, China). SiHa cells transfected with an empty lentiviral vector were employed as the control group (Lv-Control). Lentiviral vectors of miR-154-5p precursor (pre-miR-154-5p) and sponge fragments were introduced into 293T cells, packaged with lentivirus, and infected with SiHa cell after purification of the lentivirus solutions. After 48 h, puromycin (Solarbio, Beijing, China) at a concentration of 3.0 µg/mL was added to screen stable overexpressed and silenced cell lines, named Lv-miR-154-5p-OE group and Lv-miR-154-5p-Sp group, respectively. Real-time quantitative polymerase chain reaction (RT-qPCR) was utilized to verify miR-154-5p expression in cells.

FV136 lentiviral vector (Fubio, Suzhou, China) with CUL2 overexpression was prepared and infected into SiHa cells of the Lv-miR-154-5p-OE group. Screening drug hygromycin (Fubio, Suzhou, China) at 300 µg/mL concentration was added to obtain a SiHa stable cell line model of miR-154-5p and CUL2 co-overexpression, named Lv-CUL2-OE group. SiHa cells of the Lv-miR-154-5p-OE group transfected with the empty sequence vector were used as a control group (Lv-CUL2-CO). The expression levels of CUL2 mRNA and CUL2 proteins in the stable cell lines were verified by RT-qPCR and Western blot, respectively.

Real-time quantitative polymerase chain reaction

Overall RNA was extracted from samples using Trizol lysate. A260/A280 ratio of 1.9–2.2 was used to confirm the quality and purity of the RNA, which was reverse transcribed to cDNA. RT-qPCR was employed to measure the mRNA expression using real-time fluorescent quantitative universal reagents (Cat#GMRS-001; GenePharma, Shanghai, China) and real-time fluorescent quantitative PCR instrument (FTC-3000, Funglyn Biotech, Canada). We conducted quantitative PCR reaction program as follows: denaturation at 95 °C for 3 min, followed by 40 cycles of 95 °C for 12 s, 62 °C for 30 s, 72 °C for 30 s. Glyceraldehyde-3-phosphate dehydrogenase (GAPDH) and U6 were employed as endogenous reference genes, and the specific primer sequences were designed and synthesized by Shanghai GenePharma Company. Every gene expression was computed applying the 2−∆∆Ct function. The primer sequences of RT-qPCR used in this investigation are as follows:

miR-154-5p forward primer 5′-CGCGTAGGTTATCCGTGTTG-3′ and reverse primer

5′-AGTGCAGGGTCCGAGGTATT-3′;

U6 forward primer 5′-GGAACGATACAGAGAAGATTAGC-3′ and reverse primer

5′-TGGAACGCTTCACGAATTTGCG-3′;

CUL2 forward primer 5′-TATGTGTGGCCTATCCTGAACC-3′ and reverse primer

5′-TGCAAATGCCGAACATGATTTTC-3′;

GAPDH forward primer 5′-GGAGCGAGATCCCTCCAAAAT-3′ and reverse primer

5′-GGCTGTTGTCATACTTCTCATGG-3′.

CCK-8 assay

A 96-well plate was used for the stable SiHa cells growing at a density of 3 × 103 cells per well, and the plate was removed after pre-culture for 24 h. Then, 10 μL CCK-8 (Absin, Shanghai, China) was put into each well under dark conditions and 1 h in an incubator. We monitored the OD450 value at five time-points (0, 24, 48, 72, and 96 h) using a microplate reader.

Cell cycle distribution

The cells were digested with trypsin without EDTA. The collected cells (1.0 × 107) were rinsed twice with PBS, then with pre-cooled 75% ethanol diluted in PBS, and regularized at −20 °C overnight. A 5 min centrifugation was performed on cells at 1,000 g, discarding the supernatant, the cell pellet was rinsed with PBS, and centrifuged again. Next, 500 μL prepared propidium iodide staining solution (Yeasen, Shanghai, China) (supplemented 10 μL propidium iodide stock solution and 10 μL RNaseA to 500 μL staining buffer) was added to cell samples to resuspend the cells. Cell samples were incubated in a 37 °C water bath (Yiheng Instruments Co., Ltd., Shanghai, China) without light for 30 min, and the NovoCyte flow cytometer (ACEA Pharma, Hangzhou, China) was used for detection.

Colony formation assay

Cells were inoculated in a six-well plate at a density of 1,000 cells per well and cultured in an incubator for 2 weeks. The medium was discarded, and every well of the culture plate was gently rinsed twice with PBS and fastened with 4% paraformaldehyde (Meilunbio, Dalian, China) for 20 min and stained with 0.5% crystal violet (Solarbio, Beijing, China) for 20 min. After staining was completed, the colonies were scanned and imaged by 3D HISTECH scanner (Budapest, Hungary).

Would healing assay

The stable SiHa cells were inoculated into a six-well plate during logarithmic growth phase, and when the cell fusion rate reached approximately 80%, a 200 μL pipette tip was employed to streak it vertically. The scratched cells were washed three times gently with sterile PBS, and serum-free medium was supplemented for culture. Images were acquired in the same region of view at 0 and 48 h with an inverted microscope (Leica DMIL LED, Weitzlar, Germany), and the cell migration index was calculated to evaluate cell migration ability. Cell migration index = (0 h scratch void region − 48 h scratch void region)/0 h scratch void region.

Bioinformatic prediction of targets and potential functions of miR-154-5p

Datasets of downstream target genes of miR-154-5p were downloaded from three online tools: TargetScan (https://www.targetscan.org/vert_72/) (Release 7.2, 21 July, 2021 accessed), The Encyclopedia of RNA Interactomes (Encori) (https://starbase.sysu.edu.cn/) (21 July, 2021 accessed) and miRDB (https://mirbase.org/) (21 July, 2021 accessed) (Li et al., 2022). The overlapping genes from the above databases were managed by Venny 2.1.0 (Chen et al., 2022) (https://bioinfogp.cnb.csic.es/tools/venny/) and uploaded to the STRING database (https://www.string-db.org/) (Version 11.5, 1 August, 2021 accessed) to observe protein-protein interactions (Szklarczyk et al., 2021). Meanwhile, the pathway enrichment and functional clustering of the downstream overlapping genes of miR-154-5p were analyzed and visualized based on the Kobas 3.0 website (http://kobas.cbi.pku.edu.cn/) (17 October, 2021 accessed) (Bu et al., 2021). For each group set, the first five terms with the highest enrichment ratio were shown if there were more than five terms. Based on the above analysis, the potential biological effect and target mRNAs of miR-154-5p were further understood.

Access to the open data

Data on the pan-cancer expression profile of CUL2 (Ensembl ID: ENSG00000108094.14) were retrieved from the Gene Expression Profiling Interactive Analysis (GEPIA) database (http://gepia.cancer-pku.cn/) (17 October, 2021 accessed) (Zhang et al., 2023).

Xenograft mouse model

A total of 18 nude mice were chosen, randomly divided into three groups, with six in each group. SiHa stable cell lines in Lv-Control, Lv-miR-154-5p-OE and Lv-miR-154-5p-Sp groups were made into a single cell suspension (PBS: Matrigel = 7:3) and injected subcutaneously into mice at an inoculation density of 1.0 × 107. At 7 days, the modeling was considered successful if the grafted tumor became visible and hard to touch. Tumor growth was observed, the length and width of graft tumor were measured every 4 days, and a curve was drawn. On day 49, we terminated the animal experiment after a clear trend of observable tumor growth sufficient for RNA and protein extraction. The mice were euthanized quickly via cervical dislocation. Tumors were then isolated, weighed, and photographed. Calculation formula for tumor volume: V (volume) = (length × width2)/2 (Zhang et al., 2022a).

Moreover, six additional nude mice were obtained and divided into two groups, with three in each group. Cell suspensions from Lv-CUL2-CO group and Lv-CUL2-OE group were injected into these mice to perform the tumor xenograft assay. Tumor cells were successfully modeled at 7 days, and nude mice in the CUL2-OE and CUL2-CO groups were euthanized at 31 days.

In vivo metastasis assay

We obtained eight additional mice and randomly divided them into four groups: blank group (no cell injection) (n = 1), control group (Lv-Control) (n = 1), miR-154-5p overexpression group (Lv-miR-154-5p-OE) (n = 3), and miR-154-5p sponge group (Lv-miR-154-5p-Sp) (n = 3). They were fixed in a visual tail vein injection fixture (Yiyan, Jinan, China), using an alcohol cotton ball to dilate the vein of the mouse tail by a temperature difference. A 1 mL syringe was used to inject 2 × 106 GFP-labeled SiHa cells (100 μL) into the tail vein. The mice were observed to exclude abnormal states, after which they were kept under SPF conditions. The tumor metastasis ability and extent of SiHa cells in vivo were evaluated. After 60 days, the mice were euthanized, and the liver, spleen, lung, kidney, and uterus were dissected. Then, the GFP signal intensity in each organ was precisely surveyed using the IVIS Spectrum small animal in vivo optical image system (PerkinElmer, Waltham, MA, USA) (Chen et al., 2021).

Western blot

The RIPA lysate (Boster, Wuhan, China) was mixed with protease inhibitors (Boster, Wuhan, China), and animal tumor tissues were lysed and total protein was extracted. The BCA concentration determination kit (Boster, Wuhan, China) detected the protein concentration and determined the loading amount (cell samples: 25 µg, animal samples: 30 µg). SDS-PAGE gel (8%) for electrophoresis was prepared and the protein was transferred to the polyvinylidence fluoride (PVDF), which was sealed with 5% degreased milk powder for 2.5 h. The primary antibodies against CUL2 (Cat# sc-166506, 1:100; Santa Cruz, CA, USA) and β-actin (Cat# 4970, 1:1,000, CST, Boston, MA, USA) were added and cultured overnight in a refrigerator at 4 °C. Then, the secondary antibody was added and the sample was incubated for 1 h. The ECL chemiluminescent solution (Seven, Beijng, China) was used for visualization via Gel imaging system (ChemiDocTM XRS; Bio-Rad, Hercules, CA, USA). Using β-actin as the internal reference control, the correlative expression of CUL2 protein was computed as the quotient of gray value of CUL2 and β-actin via Image Lab 5.2 software.

Hematoxylin and eosin (HE) staining

The dissected tumor samples of the nude mice were fastened with formaldehyde (Meilunbio, Dalian, China), dehydrated, embedded, and sectioned. After HE staining, we used Adobe Photoshop CS6 to monitor the presence of cancerous foci.

Immunohistochemical analysis

After the mice tumor tissues were dehydrated, waxed, embedded, and cut into sections, each section of tissue was deparaffinized in xylene and hydrated using ethanol at different concentration gradients. High pressure was used for antigen retrieval and blocked with goat serum, and CUL2 antibody (Cat#51-1800, 1:200; Invitrogen, Carlsbad, CA, USA) was subjoined and cultured overnight at 4 °C. Secondary antibody was subjoined after PBS rinsing. The secondary antibody was removed, rinsed with PBS, stained with freshly prepared DAB staining solution, and the degree of staining was monitored under a microscope (Leica DM500; Leica, Weitzlar, Germany). After counterstaining with hematoxylin and differentiation with hydrochloric acid and alcohol, tap water was used to reverse the blue color. After dehydration with different concentrations of ethanol and transparent xylene, the slides were mounted with neutral gum. Under electron microscopy, a cytoplasm or nucleus exhibiting brownish yellow coloration was regarded as a positive finding. Five regions of view were randomly chosen for analysis under a 400× field of view. Using Adobe Photoshop (version: CS6) and Image pro (version: plus 6.0), the average value of the integrated optical density (IOD) of the five fields of view was taken as the statistic.

Statistical analysis

SPSS 25.0 and GraphPad Prism (version 9.0) software packages were used for data processing and chart drawing. Data from the research were shown as the mean ± standard deviation (SD). Count data are shown as n. The student t-test was conducted for the comparison of the two groups. One-way analysis of variance was used to compare the three or more groups. P value < 0.05 was considered statistically significant, and all experiments were replicated independently in triplicates.

Results

Construction of SiHa cell lines with stable high and low miR-154-5p expression

To evaluate the effect of miR-154-5p on cervical cancer cells, we first performed RT-qPCR on SiHa and HaCaT cells to detect expression levels of miR-154-5p. The results indicated its expression level was lower in SiHa cells than in HaCaT cells (Fig. 1A). Subsequently, we created SiHa stable cell lines with overexpressed and silenced miR-154-5p, named Lv-miR-154-5p-OE and Lv-miR-154-5p-Sp, respectively. Meanwhile, the SiHa cells transfected with an empty lentiviral vector were employed as the control group (Lv-Control) (Fig. 1B). To ensure that cell models could be used in subsequent experiments, RT-qPCR was used for verification. As shown in Fig. 1C, compared with the Lv-Control group, miR-154-5p expression in the Lv-miR-154-5p-OE group was significantly increased, but there was no difference in the Lv-miR-154-5p-Sp group.

Figure 1 Construction of SiHa cell lines with stable high and low miR-154-5p expression.

Low expression of miR-154-5p in SiHa cells and the efficiency of lentivirus-mediated miR-154-5p overexpression or silencing in SiHa cells. (A) miR-154-5p expression level in SiHa (HPV16-positive human cervical cancer cell lines) and HaCaT (Human immortalized epidermal cell lines) was assayed via RT-qPCR; **, P < 0.01. (B) GFP expression was observed in the infected SiHa cells by fluorescence and optical microscopy (100×). Representative images were selected; BL, brightfield; FL, fluorescence. (C) RT-qPCR was used to verify the construction effect of SiHa cell lines with stable high and low miR-154-5p expression; ***, P < 0.001 (Lv-miR-154-5p-OE vs Lv-Control). Experiments in (A) and (C) were analyzed using two-tailed student’s t-test and one-way ANOVA, respectively; they were repeated three times, including three technical replicates and three biological replicates. Error bars represent mean ± SD. ANOVA, analysis of variance; GFP, green fluorescent protein; RT-qPCR, real-time quantitative polymerase chain reaction; SD, standard deviation.

Effect of miR-154-5p on the proliferation and migration of SiHa cells

CCK-8, colony formation, and cell cycle measurements were used to assess the effects of miR-154-5p on the proliferation of SiHa cells. The overexpression group resulted in weaker cell proliferation and lower malignancy rates in comparison to the Lv-Control group, and the proportion of cells in the G1 phase was increased. In contrast, silencing miR-154-5p boosted the proliferation of SiHa cells, enhanced cell colony formation ability, and lowered the scale of cells in the G1 phase (Figs. 2A–2C). Immediately afterward, we assessed the migration capabilities of SiHa cells by scratch assay. As shown in Fig. 2D, miR-154-5p overexpression restrained the migration of SiHa cells, and its knockdown promoted their migration, in comparison to the Lv-Control group.

Figure 2 Effect of miR-154-5p on the proliferation and migration of SiHa cells.

Influences of miR-154-5p overexpression and silencing on proliferation and migration of SiHa cells. (A) CCK-8 assay was performed to detect cell viability at different time points; *, P < 0.05; **, P < 0.01; ***, P < 0.001. (B) Clonogenic assay was performed to evaluate the colony-forming ability of SiHa cells; ***, P < 0.001. (C) Typical graphs and cell population of cell cycle distribution in different groups, measured by flow cytometry using the PI staining method; *, P < 0.05; **, P < 0.01; ***, P < 0.001. (D) Typical photos indicate the initial (0 h) and final (48 h) positions of cells after scraping from various groups via the wound healing assay (100×); **, P < 0.01; ***, P < 0.001. Experiments in (A)-(D) were replicated independently in triplicates, including three technical replicates and three biological replicates. Error bars represent mean ± SD. Data were analyzed by one-way ANOVA. ANOVA, analysis of variance; PI, propidium iodide; SD, standard deviation.

CUL2 was associated with miR-154-5p and was highly expressed in cervical cancer

To further elucidate the mechanism of miR-154-5p in the tumorigenesis of cervical cancer, we screened the downstream target genes of miR-154-5p and elucidated its function, as shown in Fig. 3.

Figure 3 Flow chart of target mRNAs screening and functional annotation for miR-154-5p.

The three databases (TargetScan, Encori, and miRDB) showed 169, 2,097, and 339 genes, respectively, with targeted binding to miR-154-5p. According to Venny 2.1.0 analysis, a total of 38 genes overlapped in the above database (Fig. 4A). The dataset of genes in the overlapping regions was uploaded to the STRING database to retrieve the interaction relationship between proteins. Homo sapiens, full STRING network, medium confidence (0.400), and FDR stringency (five percent) were selected as the filtering criteria by default. Of 37 other proteins, CUL2 interacted with only three proteins (UBE2H, FBXL16, and SOCS5) (Fig. 4B). Meanwhile, Kobas 3.0 was utilized to perform pathway enrichment and functional clustering analysis of 38 genes. The visual bubble diagram showed that the downstream overlapping genes of miR-154-5p were co-clustered into seven categories (C1–C7): C1 referred to target genes mainly distributed in intracellular components, C2 encompassed protein ubiquitination-related pathways, C3 represented 38 genes with multiple protein kinase activities and protein binding functions, C4 represented congenital disorders, C5 encompassed solid tumor signaling pathways (including renal cell carcinoma and glioma), C6 included glucagon and oxytocin signaling pathways, and C7 included signaling pathways related to metabolism (Fig. 4C). These findings show that miR-154-5p alters certain tumor-related biological processes by modulating the 38 genes.

Figure 4 miR-154-5p target mRNAs screening and functional annotation.

(A) The Venny 2.1.0 analysis of three database sets (TargetScan, Encori, and miRDB). (B) Protein interaction map of 38 genes. (C) Kobas 3.0. was employed to analyze the pathway enrichment and functional clustering of the downstream overlapping genes of miR-154-5p.

Next, we searched the pan-cancer expression profile of CUL2 using the online database GEPIA and discovered that CUL2 expression was higher in CESC (cervical cancer, including cervical squamous cell carcinoma and adenocarcinoma) than in normal cervical tissue (Fig. 5A). Subsequently, we detected the level of CUL2 in SiHa cell, HaCaT cell, Lv-Control, Lv-miR-154-5p-OE and Lv-miR-154-5p-Sp SiHa stable cell lines by RT-qPCR. CUL2 mRNA expression was higher in SiHa cells than in HaCaT cells (Fig. 5B). Moreover, in SiHa stable cell lines, compared with the Lv-Control group, the Lv-miR-154-5p-OE group showed reduced CUL2 mRNA expression, while the Lv-miR-154-5p-Sp group showed no statistically significant difference in its expression (Fig. 5C).

Figure 5 CUL2 was highly expressed in cervical cancer.

(A) Pan-cancer expression profile of CUL2 was displayed by accessing the GEPIA database. (B) CUL2 mRNA expression in SiHa cells and HaCaT cells was quantified via RT-qPCR; **, P < 0.01. (C) The CUL2 mRNA expression was detected in the SiHa cells of Lv-miR-154-5p-OE/Sp group and the Lv-Control group; ***, P < 0.001 (Lv-miR-154-5p-OE vs Lv-Control). Experiments in (B) and (C) were repeated three times, including three technical replicates and three biological replicates. Error bars represent mean ± SD. Data were analyzed using two-tailed student’s t-test and one-way ANOVA, respectively. ANOVA, analysis of variance; RT-qPCR, real-time quantitative polymerase chain reaction; SD, standard deviation.

MiR-154-5p hindered the growth and metastasis of cervical cancer by targeting CUL2 in vivo

To study whether miR-154-5p was capable of hindering the development and metastasis of SiHa cervical cancer cells in vivo, mice (n = 18) were injected subcutaneously with Lv-Control and Lv-miR-154-5p-OE/Sp cell suspension, and the tumor weight and size were measured. The average tumor volume and weight in the overexpression group were smaller than in the Lv-Control group (Figs. 6A and 6B). RT-qPCR showed that the overexpression group had higher miR-154-5p expression and lower CUL2 mRNA (Figs. 6C and 6D). Western blot and IHC revealed that CUL2 expression in the overexpression group was downregulated compared to the control group, while this phenomenon was opposite in the silenced group (Figs. 6E and 6F).

Figure 6 miR-154-5p suppressed the growth of cervical carcinoma by targeting CUL2 in vivo.

(A) The tumor volume of animals in each group was measured every 4 days; **, P < 0.01 (Lv-miR-154-5p-OE vs Lv-Control). (B) Representative photos of cervical cancer xenograft tumor and the weight of the tumor in each animal were recorded; **, P < 0.01 (Lv-miR-154-5p-OE vs Lv-Control). (C) miR-154-5p in tumor tissues in mice was detected using RT-qPCR; *, P < 0.05 (Lv-miR-154-5p-OE vs Lv-Control). (D) CUL2 mRNA in tumor tissues in mice was detected using RT-qPCR; **, P < 0.01 (Lv-miR-154-5p-OE vs Lv-Control). (E) The relative expressions of CUL2 in mice tumor tissues were quantified using western blot via normalization to β-actin; *, P < 0.05. (F) HE and IHC staining of tumors in indicated groups. n = 6. Experiments in (C)-(F) were repeated three times, including three technical replicates and three biological replicates. Error bars represent mean ± SD. Data were analyzed by two-tailed student’s t-test. HE, hematoxylin-eosin; IHC, immunohistochemistry; RT-qPCR, real-time quantitative polymerase chain reaction; SD, standard deviation.

In the tail vein injection model, we used an in vivo imaging instrument to observe the metastasizing signal intensity in cancer cells in mice. The liver, spleen, lung, kidney, and uterus of the mice showed positive signals compared to the Lv-Control group. Except for the liver, spleen, and kidney, the mean values of the radiant efficiency in the lung and uterus of mice in the Lv-miR-154-5p-OE group were significantly decreased (Figs. 7A and 7B).

Figure 7 miR-154-5p suppressed the metastasis of cervical carcinoma by targeting CUL2 in vivo.

(A, B) Metastasis in major organs (liver, spleen, lung, kidney, and uterus) was assayed using bioluminescent imaging and GFP filters of each organ; *, P < 0.05. Error bars represent mean ± SD. Data were analyzed by one-way ANOVA. ANOVA, analysis of variance; GFP, green fluorescent protein; SD, standard deviation.

Overexpression of CUL2 influenced the effect of miR-154-5p on cervical cancer

To further demonstrate whether the effects of miR-154-5p on cervical cancer were influenced by CUL2, CUL2 was stably overexpressed in the background of the Lv-miR-154-5p-OE group, named Lv-CUL2-OE, and miR-154-5p-overexpressing SiHa cells transfected with an empty vector were considered the control group (Lv-CUL2-CO) (Fig. 8A). They were then verified by RT-qPCR and western blot, the outcomes illustrated that the expression levels of CUL2 mRNA and CUL2 protein were all higher in the Lv-CUL2-OE group than in the Lv-CUL2-CO group (Figs. 8B and 8C). Further, in comparison to the Lv-CUL2-CO group, the Lv-CUL2-OE boosted the proliferation and cloning capability, lowered the proportion of cells in the G1 phase, and promoted the migration capability of SiHa cells (Figs. 8D–8G).

Figure 8 Overexpression of CUL2 influenced the effect of miR-154-5p on cervical carcinoma.

(A) GFP expression was observed in the infected SiHa cells by fluorescence and optical microscopy (100×); BL, brightfield; FL, fluorescence. (B) RT-qPCR detected CUL2 mRNA expression in SiHa cells of Lv-CUL2-OE group and Lv-CUL2-CO group using corresponding primers; *, P < 0.05. (C) Western blot analysis and quantitative analysis showed the CUL2 protein with the transfection of both miR-154-5p and CUL2 in SiHa cells; *, P < 0.05. (D) CCK-8 assay evaluated cell proliferation ability in the SiHa cells of Lv-CUL2-OE group and Lv-CUL2-CO group; ***, P < 0.001. (E) Colony proliferation assay was conducted to confirm the clone forming ability of CUL2 overexpression in SiHa cells; ***, P < 0.001. (F) Cell cycle distribution was performed to examine the cell population in each cell cycle stage (%); **, P < 0.01; ***, P < 0.001. (G) The migration of SiHa cells was assessed after CUL2 overexpression at 48 h via wound healing assay; *, P < 0.05. (H) Representative photos of cervical cancer xenograft tumor and the weight of the tumor in each animal were recorded for the Lv-CUL2-OE group and Lv-CUL2-CO group. n = 3; *, P < 0.05. (B)-(G) was replicated independently in triplicates, including three technical replicates and three biological replicates. Error bars represent mean ± SD. Data were analyzed by two-tailed student’s t-test. GFP, green fluorescent protein; RT-qPCR, real-time quantitative polymerase chain reaction; SD, standard deviation.

In the in vivo experiment, after successful modeling of nude mice, the mice were killed 31 days after inoculation, and the weight of nude mice transplanted tumors in the CUL2-OE group was greater than that in the CUL2-CO group (Fig. 8H).

Discussion

Cervical cancer is a disease mainly caused by HPV infection and has the highest mortality rate among women worldwide. Blocking the key molecules in the pathogenesis of cervical cancer triggered by HPV infection can help optimize the treatment strategy and develop new targeted therapy drugs (Fleischmann et al., 2021).

In the process of HPV infection and cervical cancer progression, the epigenetic mechanism of miRNA regulation of related genes has come into focus. Low expression of miR-199a could block the inhibitory effect of HDAC6 silencing on the development of cervical cancer cells in vivo and in vitro (Shao et al., 2021). The epigenetic adjustment of miR-142-5p inactivated the PI3K/AKT signaling pathway by targeting PIK3AP1 and inhibiting cervical cancer progression (Guo et al., 2021). In our study, we focused on miR-154-5p of the miRNA expression profile in cervical cancer and found that it was remarkably low expression in cervical cancer cells, leading to the speculation that it is a tumor suppressor miRNA with an important impact on cervical cancer. Of note, this study was an extension and refinement of our previous work. First, in terms of research design, bioinformatics combined with molecular biology experiments were used to elucidate the important role of miR-154-5p in the development of cervical cancer. Second, in terms of experimental technology, we have repeatedly used lentivirus vector tools to construct stable cell lines. Finally, in terms of research content, stable cell lines were inoculated into the subcutaneous and tail vein of nude mice, adding in vivo experiments that dynamically monitor the progression of cervical cancer.

MiRNA sponge is a type of mRNA molecule that serves as a functional unit for biological processes. It acts as a sponge for absorbing miRNAs, thus separating them from their targets and inhibiting their function (Zhang et al., 2022b). In addition to miR-154-5p overexpression, we designed the sponge sequence to observe the biological features of cervical cancer through the differential expression of miR-154-5p. As mentioned above, miR-154-5p overexpression remarkably hindered the proliferation, migration, and colony formation ability of SiHa cells and concomitantly led to G1 arrest of the cell cycle. Although miR-154-5p-Sponge did not significantly downregulate the miR-154-5p level, the introduction of sponge fragments blocked the effect of miR-154-5p by absorbing miR-154-5p. We observed the phenomenon that silencing miR-154-5p promotes the proliferation, migration, and colony forming ability of SiHa cells and reduces the proportion of cells in the G1 phase. Thus, we inferred that this is caused by the unique mechanism of “sponge,” which binds competitively rather than causing complete degradation.

CUL2 has a significant role in the development and prognosis of cervical cancer and other malignant tumors (Chu et al., 2021). Xu et al. (2016) confirmed that CUL2 is recruited by E7 oncoprotein, which is driven by the CUL2/E2F1/miR-424 regulatory loop, upregulating CUL2 and accelerating the occurrence of HPV16-induced cervical cancer. Here, we uploaded the datasets of downstream target mRNA of miR-154-5p from three public databases (TargetScan, Encori, and miRDB) and observed that CUL2 was connected with three proteins in the protein interaction network. Further search of the pan-cancer expression profile of CUL2 revealed that its expression was significantly higher in patients with uterine tumors than in normal patients. At the cellular level, we also found that CUL2 mRNA expression was higher in SiHa cells compared to HaCaT cells. The above evidence suggests that the high expression of CUL2 may be related to the occurrence of cervical cancer.

Interestingly, we found that over-expression of miR-154-5p downregulated CUL2 mRNA expression levels in SiHa cells, but its down-expression had no obvious effect on the level of CUL2 mRNA. The reason for this phenomenon could be that the effect of miR-154-5p-targeted inhibition of CUL2 occurs at the post-transcriptional level. It also suggested that the inhibitory effect of miR-154-5p on SiHa cell proliferation and migration may be achieved by reducing the expression of CUL2. However, in vivo evidence on whether miR-154-5p targets CUL2 to regulate cervical cancer is still lacking in the literature.

Lentiviral vectors can target cells that are difficult to transfect and integrate the gene payload into the host genome (Gopal et al., 2021; Abu Halim et al., 2021). Here, we integrated the gene sequences of pre-miR-154-5p, miR-154-5p sponge, and CUL2 into lentiviral vectors to achieve stable overexpression of miR-154-5p and CUL2 or stable interference of miR-154-5p. This provided a great technical advantage for establishing animal models, avoiding the defects of long cycles and poor stability of animal models. In this study, the tumor tissues were dissected, aliquoted in tissue tubes, and stored at −80 °C to detect miR-154-5p and CUL2 expression levels to assess the effects of miR-154-5p on the occurrence and development of cervical cancer in vivo. These outcomes revealed that miR-154-5p upregulation led to reduced weight and volume of tumors in nude mice and decreased CUL2 expression, while its downregulation improved CUL2 protein level but had no obvious influence on tumor weight and volume and CUL2 mRNA level, which demonstrated that miR-154-5p inhibited CUL2 translation via silencing CUL2 in vivo, thereby hindering the progression of cervical cancer. Besides, in vitro and in vivo experiments of Fig. 8 demonstrated that the co-overexpression of CUL2 and miR-154-5p influenced the effect of miR-154-5p overexpression in cervical carcinoma, including promoting proliferation, migration, and tumor growth of cervical cancer. Thus, the tumor suppressor effects of miR-154-5p on cervical cancer might have been altered by the expression of CUL2. However, we lacked evidence for a functional recovery experiment with or without overexpression of CUL2 in SiHa cells transfected with control or miR-154-5p overexpression/inhibitor to thoroughly analyze cervical carcinogenesis.

Metastasis is a primary driving factor for the adverse outcomes of cervical cancer (Li et al., 2021b). Traditionally designed studies on tumor metastasis use HE and immunohistochemistry staining to count the number of metastatic nodules. However, relying only on the detection and verification of cancer cell behavior cannot intuitively, dynamically, and comprehensively measure and evaluate the growth and outcome of tumors. Biological processes in vivo help in understanding and clarifying the progression of tumors. Therefore, we used the method of tail vein injection combined with small animal live imaging to simulate and evaluate the occurrence and development of tumor metastasis in vivo and objectively and accurately evaluated the ability of tumor metastasis. The results showed that SiHa cells had metastasized dramatically in menstrual blood 60 days after injection. Compared to the Lv-Control group, the average radiant efficiency of the lung and uterus in the treated group with miR-154-5p overexpression decreased significantly. These results suggest that miR-154-5p slowed down the metastasis of cervical cancer cells in the body.

Conclusions

Overall, this study comprehensively revealed that miR-154-5p silenced CUL2 expression and inhibited translation, thus weakening the proliferation and migration capabilities of HPV16-positive cervical cancer cells, ultimately inhibiting cervical carcinoma growth and metastasis. However, the effect of miR-154-5p sponge on tumor growth and metastasis in animals is not obvious, which may be related to sponge degradation in vivo, and needs further research and investigation. In addition, the potential feedback loop provided by miR-154-5p is large, and its upstream regulatory mechanisms in cervical cancer should be explored in depth.

Supplemental Information

Supplemental Information 1 Raw data.

Click here for additional data file.

Supplemental Information 2 Raw data for Figure 8D.

CCK-8 assay

Click here for additional data file.

Supplemental Information 3 ARRIVE 2.0 Checklist.

Click here for additional data file.

Supplemental Information 4 Uncropped Blots.

Click here for additional data file.

Additional Information and Declarations

Competing Interests

Author Contributions

Ethics

Data Availability

Fubio Biotechnology Co., Ltd (Suzhou, China) provided experimental technical support and assistance. The authors declare that they have no competing interests.

Yaqin Li conceived and designed the experiments, performed the experiments, prepared figures and/or tables, authored or reviewed drafts of the article, and approved the final draft.

Yimiao Wei performed the experiments, analyzed the data, prepared figures and/or tables, and approved the final draft.

Honglei Zhang performed the experiments, analyzed the data, prepared figures and/or tables, and approved the final draft.

Ying Bai analyzed the data, prepared figures and/or tables, and approved the final draft.

Xiuting Wang analyzed the data, prepared figures and/or tables, and approved the final draft.

Qi Li analyzed the data, prepared figures and/or tables, and approved the final draft.

Yatao Liu analyzed the data, prepared figures and/or tables, and approved the final draft.

Shuling Wang analyzed the data, prepared figures and/or tables, project administration of this research, and approved the final draft.

Jiapu Wang analyzed the data, prepared figures and/or tables, provided the resources and supervised the work, and approved the final draft.

Songquan Wen analyzed the data, prepared figures and/or tables, project administration of this research, and approved the final draft.

Jiarong Li analyzed the data, prepared figures and/or tables, project administration of this research, and approved the final draft.

Weihong Zhao conceived and designed the experiments, prepared figures and/or tables, authored or reviewed drafts of the article, and approved the final draft.

The following information was supplied relating to ethical approvals (i.e., approving body and any reference numbers):

Animal experiment was approved by the Standards for Ethics committee of the Second Hospital of Shanxi Medical University (No. DW2022035) and performed in accordance with the relevant experimental animal guidelines and regulations for the care and use of animals.

The following information was supplied regarding data availability:

The raw data are available in the Supplemental Files.

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
