# Peer review of "MicroRNA-154-5p suppresses cervical carcinoma growth and metastasis by silencing Cullin2 in vitro and in vivo"

_PeerJ, doi:10.7717/peerj.15641_

## Round 0.1 · original submission · Major Revisions

Dear Dr. Zhao,

This manuscript has been reviewed by two reviewers. Although the work is interesting, the reviewers have raised several concerns. In addition to the comments of the reviewers, the authors may address the following points in the revised manuscript.

1. This article is an extension of their own work (PMID: 32742484; doi: 10.7150/jca.45871) published in 2020. The authors should discuss the similarity and advances in the current article compared to their previous publication in the discussion section.

2. It would be better to describe the logic of choosing miR-154 for the investigation.

3. The authors have performed individual experiments with miR-154 overexpression/inhibition and CUL2 overexpression. It might be better to perform a rescue experiment with or without overexpression of CUL2 in cells transfected with control or miR-154 inhibitor; and analyze cervical carcinogenesis.

Look forward to receiving the revised manuscript soon.

Reviewer 1 ·

Basic reporting

The overall language in the main text is clear and professional. The figures are displayed nicely and raw data was shared. However, the figure captions were sometimes confusing and could be improved. Below are several potential points for improvement.

1. The introduction to the relevance of miRNA in HPV-induced tumorigenesis was good. However, it would be helpful to include more molecular-level details about how miRNAs work, including their involvement in RNA silencing and post-transcriptional regulation of gene expression, etc.

2. Line 266: “The results indicated its expression level in SiHa cells was downregulated, compared to HaCaT cells.” It would be more appropriate to say that “the expression level is lower in SiHa cells compared to HaCaT cells” instead of saying that it is downregulated unless there is a specific downregulation event.

3. For all plots with error bars, the number of replicates and what the error bars represent should be clearly indicated in the figure captions.

4. Here are some suggestions that may help enhance the readability of the figure captions. The figure caption should include a concise description of the experiment and any additional information needed to interpret the data, including but not limited to what cell line, what the abbreviations/symbols mean, the number of replicates performed, what tests were done, what error bars represent, etc. Some figure captions were declarative and stated the main finding (i.e., Figure 1A, 4A, 4C, 5B, 8D); this was a bit repetitive from the main text and may be unnecessary. An alternative caption that is more descriptive of the experiment may be more appropriate.

For example, in Figure 1, the caption of Figure 1A states, “Compared with HaCaT, the miR-154-5p expression level in SiHa was significantly reduced” – this was already stated in the main text line 265-266. An alternative caption describing the RT-qPCR experiment, including what the qPCRs were normalized to, what the error bars represent, and what the “**” represent, is more appropriate. Figure 1B: the FB and FL captions could be moved in line with the existing caption for 1B to avoid confusion. Figure 1C: for the “**, p<0.01; ***, p<0.001”, it should indicate what test was used. Student’s t-test / One vs. Two-tailed? The captions also did not describe what the error bars on 1A and 1C represent. How many biological or technical replicates were performed? This is just an example from Figure 1. The other figure captions should be adjusted accordingly.

5. Figure 5 caption is found in Figure 4 caption section.

6. Figure 5C – the cell line wasn’t indicated in the main text or figure caption.

Experimental design

The experimental design was appropriate. The research question was well-defined. Experimental methods were described with sufficient detail.

Bioinformatic prediction methods should add more detail, including release/version and/or date accessed. STRING Database search/mapping settings were not clear in Figure 4B – information about the selected interaction score (confidence), physical subnetwork vs. full string network should be indicated.

The rationale for a STRING Database and the Kobas 3.0 pathway enrichment and functional clustering analysis was not clear. The inputs into these searches were target genes of miR-154-5p identified to be overlapping in these databases. Looking at the interactome and the pathways these genes are involved in could give more insight into the pathways that could be affected by miR-154-5p but does not give justification for the selection of CUL2 as the protein of interest. The interpretation from these analyses should be adjusted. More will be discussed in the “validity of findings” section.

Validity of the findings

The bioinformatics analysis identifying CUL2 as a target gene of miR-154-5p is clearly established and overlaps across multiple databases, but the STRING Database and the Kobas 3.0 pathway enrichment and functional clustering analysis on the identified target genes were misinterpreted. Biological validation experiments of CUL2 as a target of miR-154-5p were convincing, and the in vivo data was impressive and demonstrated the conclusion well. However, some parts of the mechanistic association of miR-154-5p and CUL2 were slightly inadequate. Below are some comments.

1. The 38 genes were targets of miR-154-5p – they are not necessarily connected in the context of their protein-level interactions. Further, in lines 294-295: “The outcomes showed that CUL2 is a central gene in 38 genes, with its encoded proteins connected to the other three proteins”. STRING Database network map in Figure 4B does not indicate CUL2 as such. It was identified as an interacting partner with only 3/37 other proteins.

2. The descriptions of clusters on lines 297-302 were not clear, and the other clusters were not described. Also, the statement on lines 300-302 that “This was consistent with the function of CUL2 on ubiquitination modification and widely involved in malignant tumorigenesis” is not appropriate. The input genes are targets of miR-154-5p and are not relevant to CUL2. CUL2 is one of the 38 identified. By modulating these 38 genes, they cause alterations to certain cellular pathways that could affect downstream proteins or signaling pathways. The gene set enrichment analysis would provide that detail, and it would be good information to know. But since the argument infers that CUL2 is the driver for the ubiquitination modification and malignant tumorigenesis pathways observed in this analysis – this is a misinterpretation. The reason the pathways showed up is that multiple genes in the input list belong to a certain gene set for a particular pathway, not because of the individual activity or functions of CUL2. For example, CUL2 is part of the ubiquitin pathway, but so are FBXL16, UBE2H, and DCAF16, as shown in Figure 4B. The reason the ubiquitin pathway showed up in the gene set enrichment analysis is likely not because of CUL2 alone but rather the identification of multiple proteins involved in the pathway – they may or may not be relevant to CUL2.

3. Since the aim of the study, as stated in lines 71-77, is only to provide further evidence to support that miR-154-5p targets CUL2 and that this was implicated in previous work, it may not be necessary to overly emphasize the importance of CUL2. The STRING and gene set enrichment analysis is informative as to what pathways and functions may be affected but should not be used as the justification for selecting CUL2 as the protein of interest, as that would be a misinterpretation.

4. Line 302: “Thus, we identified CUL2 as a downstream target of miR-154-5p in this article” – this claim was made too early. It was only identified as a potential downstream target through the bioinformatic analysis and is pending further validation, which the manuscript describes in later sections.

5. Figure 8D: the Y-axis shows “Cell viability (fold change).” An explanation is needed for how the fold change was calculated. Further, there is uneven initial (~2x higher) cell viability in OE compared to CO at t = 0 h. Hence, the interpretation from this panel is unreliable and needs to be repeated with equal seeding.

6. For the cell viability and migration assays with CUL2 overexpression – it would be good to compare it with a control cell line (without miR-154-5p or CUL2 overexpression) to understand if there are still any gaps in cell proliferation or migration from miR-154-5p activity that isn’t compensated by CUL2 overexpression. Otherwise, language adjustments should be made to soften the language on CUL2’s involvement in these phenotypes.

Additional comments

This manuscript builds on the previously published study by Zhao et al. by offering more in vitro and in vivo validation studies. For that, I recommended toning down slightly on the process of identification of CUL2 as a target for miR-154-5p (i.e., the bioinformatic analysis) and focusing more on discussing the new data presented. How this new study improves on the previous study and understanding of the field should also be discussed in more in-depth.

Reviewer 2 ·

Basic reporting

In this study, importance of miR-154-5p in the development of cervical cancer was evaluated. It has been shown that reduced expression of miR-154-5p in cervical cancer could upregulate the expression of Cullin2 resulting degradation of Rb and progression of cell cycle. This phenomenon was validated by different in-vitro and in-vivo experiments. Following comments can be made:
1. Proper references in the methodology section should be given.
2. The discussion should be mechanistic in relation to the results.
3. The mechanism behind miR-154-5p expression with cell cycle arrest should be discussed along with its importance in cell proliferation (in-vitro/in-vivo), cell migration and metastasis.

Experimental design

1. The experiments were well designed.

Validity of the findings

The results were validated by different in-vitro and in-vivo experiments.

Additional comments

Not yet

---

## Round 0.2 · accepted · Accept

The authors have revised the manuscript as suggested by two reviewers and it is ready for publication.

Reviewer 1 ·

Basic reporting

no comment

Experimental design

no comment

Validity of the findings

no comment

Additional comments

I am pleased to see that the authors have made substantial improvements to the manuscript in this revision. All concerns raised in my previous review have been addressed satisfactorily.

Reviewer 2 ·

Basic reporting

The paper has been revised as per reviewers' suggestions .

Experimental design

The experiments were well designed and revised as per suggestions of the reviewers.

Validity of the findings

The findings were validated properly and revised accordingly.

Additional comments

None